

# Effect of 8-week frequency-specific electrical muscle stimulation combined with resistance exercise training on muscle mass, strength, and body composition in men and women: a feasibility and safety study

Mon-Chien Lee[1,2], Chin-Shan Ho[1], Yi-Ju Hsu[1], Ming-Fang Wu[1] and Chi-Chang Huang[1,3]

[1] Graduate Institute of Sports Science, National Taiwan Sport University, Taoyuan, Taiwan
[2] Center for General Education, Taipei Medical University, Taipei, Taiwan
[3] Tajen University, Pingtung, Taiwan

Corresponding author
Chi-Chang Huang,
john5523@ntsu.edu.tw

## ABSTRACT

In recent years, electrical muscle stimulation (EMS) devices have been developed as a complementary training technique that is novel, attractive, and time-saving for physical fitness and rehabilitation. While it is known that EMS training can improve muscle mass and strength, most studies have focused on the elderly or specific patient populations. The aim of this study was to investigate the effects of frequency-specific EMS combined with resistance exercise training for 8 weeks on muscle mass, strength, power, body composition, and parameters related to exercise fatigue. Additionally, we aimed to evaluate the feasibility and safety of EMS as an exercise aid to improve body composition. We recruited 14 male and 14 female subjects who were randomly assigned to two groups with gender parity (seven male and seven female/group): (1) no EMS group (age: 21.6 ± 1.7; height: 168.8 ± 11.8 cm; weight: 64.2 ± 14.4 kg) and (2) daily EMS group (age: 21.8 ± 2.0; height: 167.8 ± 9.9 cm; weight: 68.5 ± 15.5 kg). The two groups of subjects were very similar with no significant difference. Blood biochemical routine analysis was performed every 4 weeks from pre-intervention to post-intervention, and body composition, muscle strength, and explosive power were evaluated 8 weeks before and after the intervention. We also performed an exercise challenge analysis of fatigue biochemical indicators after 8 weeks of intervention. Our results showed that resistance exercise training combined with daily EMS significantly improved muscle mass ($p = 0.002$) and strength (left, $p = 0.007$; right, $p = 0.002$) and significantly reduced body fat ($p < 0.001$) than the no EMS group. However, there was no significant advantage for biochemical parameters of fatigue and lower body power. In summary, our study demonstrates that 8 weeks of continuous resistance training combined with daily upper body, lower body, and abdominal EMS training can significantly improve muscle mass and upper body muscle strength performance, as well as significantly reduce body fat percentage in healthy subjects.

## INTRODUCTION

Regular exercise training is known to promote physical and mental health, improve physiological metabolism, and reduce the risk of disease (*WHO, 2020*). Among the different types of exercise, muscle strength training plays an important role in preventing injury and improving muscle performance (*Beato et al., 2021*). Moreover, it has been found to prevent chronic diseases such as diabetes and osteoporosis by promoting muscle development and increasing caloric expenditure (*Anderson & Durstine, 2019*). Consequently, muscle strength training can lead to an increase in life expectancy and an overall improvement in the quality of life (*Rodrigues et al., 2022*). Currently, there is no alternative to exercise training that can completely replace the benefits of increased physical activity. However, for disabled individuals, frail elderly, or those who are unable to engage in sports due to congenital or acquired motor and sensory nerve damage, measures must be taken to maintain muscle mass and physical health, and to promote rehabilitation and recovery (*Kim et al., 2022*).

Previous research has shown that voluntary maximal muscle contractions can contribute to muscle growth without the use of external loads, but not all produce high levels of voluntary effort (*Counts et al., 2016*). However, inducing muscle contraction by stimulating certain muscle nerves with pulsed currents of different frequencies may have the effect of improving protein synthesis and muscle mass growth (*Qin et al., 2022*). Therefore, electrical muscle stimulation (EMS) devices developed in recent years have been considered a novel, attractive, and time-saving approach to physical fitness and rehabilitation training that complements traditional training methods (*Kemmler et al., 2018b*). EMS is a method that uses electrical impulses delivered through various forms of electrical current to electrodes on the target muscle (*Ludwig et al., 2019*). The electrical current causes involuntary muscle contraction, produces adaptations through non-selective synchronous recruitment of muscle fibers, and enables greater motor unit activation by increasing the pulse firing rate, resulting in exercise-like effects (*Seyri & Maffiuletti, 2011*). Repeated EMS sessions on muscles have been reported to increase capillary flexibility and blood flow in muscle fibers (*Kaplan et al., 2002*), resulting in improvements in body composition (*Kemmler et al., 2021*), limb muscle strength (*Granat et al., 1993*), and athletic performance (*Teschler et al., 2021*). Furthermore, EMS has the advantage of directly acting on skeletal muscle protein synthesis and preferentially activating type II skeletal muscle fibers compared to conventional techniques (*Filipovic et al., 2019*).

In recent years, EMS training has also gained popularity among healthy individuals and even competitive athletes (*Maffiuletti, 2006*). For athletes, EMS training has been shown to be effective in increasing muscle strength (*Choi, Hyon & Song, 2016*). In a study comparing 14 weeks of EMS and jumping training in soccer players, EMS was found to have a performance-enhancing effect and reduce post-exercise increases in CK activity (*Filipovic et al., 2016*). For non-athletes, topical EMS application is effective in activating superficial

abdominal muscles and increasing the cross-sectional area of the lateral abdominal wall and rectus abdominis (*Choi & Shin, 2021*). While both traditional resistance exercise training and EMS training have the potential to improve muscle mass and athletic performance, it appears that a combination of the two offers greater benefits. It has been hypothesized that resistance training combined with EMS may result in greater muscle fiber recruitment than resistance training or EMS alone, suggesting that higher stimulation and training intensities can be achieved with less perceived discomfort, thereby potentially producing greater adaptation after training (*Paillard, 2018*).

Despite substantial evidence that EMS benefits muscle performance, there is considerable variability in the findings regarding its effects on body composition and body fat (*Nishikawa et al., 2021*). On the other hand, the frequency parameter of pulse stimulation seems to have different effects and benefits on different muscle groups and different muscle fibers (*Weissenfels et al., 2018*). In addition, the safety and effectiveness of EMS are still controversial in practice, and some people think that excessive use of EMS may lead to adverse consequences (*Fernández-Elías et al., 2022*). Therefore, the aim of this study was to investigate the effects of frequency-specific EMS combined with resistance exercise training for 8 weeks on muscle mass, strength, power, body composition, and parameters related to exercise fatigue. And further evaluate the feasibility and safety of EMS as an exercise aid to improve body composition.

## MATERIALS AND METHODS

### Participant

We used the Harvard calculator (http://hedwig.mgh.harvard.edu/sample_size/size.html, accessed on 18 December 2022) to calculate the sample size, assuming a parallel design with a significance level of 0.05, a power of 0.8, and a standard deviation of the difference of 1.2. A total of 24 participants were required, and we recruited 28 participants to account for expected dropouts. We recruited healthy individuals aged between 20 and 40 years and excluded those with smoking or drinking habits, cardiovascular disease, neuromuscular disease, metabolic disease, asthma, pregnancy, and a body mass index (BMI) over 27. The study was conducted in compliance with the guidelines set forth by the Declaration of Helsinki and was reviewed and approved by the Institutional Review Board of Landseed International Hospital (Taoyuan, Taiwan; LSHIRB No. 21-034-A2). This trial was also registered at clinicaltrials.gov (accessed on 7 September 2023) as NCT06036953. All volunteers provided written informed consent after receiving a detailed explanation about the experimental procedures and content before participation.

### Experimental design

Qualified 14 male and 14 female subjects were included and randomly assigned to two groups with gender parity (seven male and seven female per group): (1) no EMS group, (2) daily EMS group (quadriceps, abdominal muscles, biceps, twice a day, 30 min each time). The mean age (no EMS group: 21.6 ± 1.7; daily EMS group: 21.8 ± 2.0), height (no EMS group: 168.8 ± 11.8 cm; daily EMS group: 167.8 ± 9.9 cm) and weight (no EMS group: 64.2 ± 14.4 kg; daily EMS group: 68.5 ± 15.5 kg) of the two groups of subjects were very similar

**Table 1 The EMS training protocol.**

| EMS training protocol | | | | | | |
|---|---|---|---|---|---|---|
| Cycle No. (unit) | 1 | 2 | 3 | 4 | 5 | 6 |
| Time (seconds) | 60 | 60 | 60 | 60 | 60 | 60 |
| Rate (Hz) | 30 | 40 | 50 | 30 | 40 | 50 |
| On time (seconds) | 4 | 4 | 4 | 3 | 3 | 3 |
| Off time (seconds) | 2 | 2 | 2 | 3 | 3 | 3 |
| Repeat (times) | 2 | 2 | 2 | 2 | 2 | 2 |
| Pluse width (µs) | 250 | 250 | 250 | 250 | 250 | 250 |

with no significant difference. All subjects underwent intervention for 8 consecutive weeks and performed resistance exercise training three times a week. Blood biochemical routine analysis was performed every 4 weeks from pre-intervention to post-intervention, and body composition, muscle strength, and explosive power were evaluated 8 weeks before and after the intervention. In addition, exercise challenges were performed after 8 weeks of intervention, and blood was collected before exercise, 30 min after exercise, and rest for 60 min after exercise to analyze fatigue biochemical indicators.

## Electric muscle stimulator (EMS)

The electric muscle stimulator (EMS) used in this study was from Funcare (Funcare Co., Ltd., Taichung, Taiwan). EMS electrodes were attached to the biceps of both hands, the abdomen, and the quadriceps of both legs sequentially. We refer to previously studies and modify the protocol (Adams, 2018; Qin et al., 2022), each part was stimulated once a day for 30 min using the parameters shown in Table 1. All subjects were trained in the protocol and received the intervention for 8 weeks.

## Resistance training (RT)

All subjects performed resistance training three times a week to strengthen their upper limbs, back, abdomen, and legs, respectively, using a pneumatic resistance training machine including Abs/Abduction 3520-HI5, Push-up/Pulldown 3120, Ab/Back 5310, Torsion Rehab 5340, Leg Extension/Curl 3530, and Leg Press 5540 (AB Hur Oy, Kokkola, Finland). Before beginning training, all subjects evaluated their own 3RM and calculated their 1RM according to the coefficient formula in previous literature, which was used as the standard for setting subsequent resistance training intensity (Brzycki, 1993). We refer to previously study and modified the protocol slightly (Krzysztofik et al., 2019; Schoenfeld et al., 2021). After a pre-exercise warm-up, all subjects performed one set of 15 repetitions at 60% of 1RM intensity in the first week and one set of 12 repetitions with resistance increased to 70% of 1RM intensity in the second week. The intensity was increased further, with subjects performing one set of 12 repetitions at 75% of 1RM in the third week and two sets of 10 repetitions at 75% of 1RM in the fourth week.

## Body composition

The InBody 570 device (In-body, Seoul, South Korea) is a bioelectrical impedance analyzer (BIA) that measures impedance at 1, 5, 50, 260, 500, and 1,000 kHz by the multi-frequency principle and was used in this study to measure body composition. After fasting for at least 8 h, the subject held the induction handle with both hands, with his arms outstretched, his torso at a 30° angle, stood on the bottom electrode, and kept still and did not speak during the measurement (as previoulsy described in *Huang et al. (2019)*).

## Maximal oxygen consumption

As previoulsy described (*Lee et al., 2021a*), all subjects were assessed $VO_{2max}$ by an automatic breathing analyzer (Vmax 29c; Sensor Medics, Yorba Linda, CA, USA) and on a treadmill (Pulsar; h/p/cosmos, Nussdorf-Traunstein, Germany). The protocol was modified by the Bruce protocol (*Bruce, Kusumi & Hosmer, 1973*), and also used a polar heart rate device to monitor heart rate (HR). From begin, the speed of the treadmill was set to 7.2 km/h and every 2 min increased by 1.8 km/h until fatigue. When the breathing exchange rate (the volume ratio of carbon dioxide produced to oxygen consumed, $VCO_2/VO_2$) was higher than 1.10 and the maximum heart rate (maximum heart rate = 220 − age) was reached, that was considered to be maximum oxygen consumption. The three highest $VO_{2max}$ peaks were averaged to obtain the $VO_{2max}$ values for the individual volunteers.

## Fatigue-associated and clinical biochemical variables

To assess fatigue-related indicators, subjects fasted for at least 8 h and underwent a fixed-intensity exercise challenge (60% $VO_{2max}$) with recovery blood sampling at designated time points, including baseline (0), post-exercise 30 min (E30), and rest for 60 min after exercise (R60). Fatigue-related assessments included lactate, ammonia ($NH_3$), and glucose. Additionally, to confirm the basic biochemical parameters and health status of the subjects during the experiment, all subjects were required to fast for 8 h and have their blood collected before, at 4 weeks, and at 8 weeks. The aspartate transaminase (AST), alanine aminotransferase (ALT), blood urea nitrogen (BUN), creatinine (CREA), uric acid (UA), and free fatty acid (FFA) were assessed. All biochemical indices were analyzed using an autoanalyzer (Hitachi 717; Hitachi, Tokyo, Japan) after serum was obtained from centrifuged blood.

## Maximum handgrip strength test

Each participant's maximum grip strength for both hands was measured using a Takei digital grip strength meter (T.K.K.5401; Takei Scientific Instruments Co., Ltd., Niigata, Japan). Before the actual test, participants were instructed to apply minimal force to the grip to ensure a comfortable and standardized gripping distance. To initiate the formal experiment, researchers randomly assigned either the dominant or non-dominant hand to start. Participants were then instructed to exert maximum force while squeezing the gripper with one hand and to alternate hands every 60 s to prevent fatigue. This alternating method was repeated three times, and the highest grip strength values for both hands were recorded as the data (*Lee et al., 2021b*).

## Countermovement jump assessment

The countermovement jump (CMJ) test is widely used to assess lower body speed, strength, and explosiveness (*Marques et al., 2014*; *Anicic et al., 2023*). During the test, participants were instructed to position their hands on their hips and stand on a Kistler force-measuring platform (9260AA; Kistler GmbH, Winterthur, Switzerland) with their feet. They were then asked to perform a squat until their knees were bent at a 90-degree angle and immediately execute a maximal vertical jump. To ensure accurate measurements, the instrument was calibrated according to each individual's weight, and CMJ data were collected at designated intervals. The parameters recorded included the rate of force development (RFD), relative peak force, and jump height. Each participant completed three repetitions of the CMJ test to obtain data (*Lee et al., 2022*).

## Statistical analysis

All the data are expressed as mean ± SD. Statistical analyses were performed using SAS 9.0 (SAS Inst., Cary, NC, USA). Multi-group comparisons were analyzed using one-way analysis of variance (ANOVA). Differences between before and after intervention were analyzed using two-way repeated-measures ANOVA with Bonferroni *post-hoc* test. Differences in the changes between before and after intervention in each subject among the two groups were analyzed using the Kruskal-Wallis test with Dunn *post-hoc* test. Statistical significance was set at $p < 0.05$.

# RESULTS

## Effect of 8 weeks EMS combined with RT on body composition

Table 2 presents the changes in body composition of the subjects. After 8 weeks of EMS combined with RT, there were no significant differences between the no EMS group and the daily EMS group in terms of body weight, BMI, muscle mass, and fat mass. However, in the no EMS group, there were significant increases in body weight ($p < 0.001$), BMI ($p < 0.001$), and fat mass ($p < 0.001$) after 8 weeks of intervention, compared to before. Conversely, in the daily EMS group, there were significant reductions in body weight ($p < 0.001$), BMI ($p < 0.001$), and fat mass ($p < 0.001$), and a significant increase in muscle mass ($p < 0.018$) after 8 weeks of intervention, compared to before.

Additionally, the changes in body composition were assessed by calculating the delta (difference) before and after the intervention. The daily EMS group exhibited a significant reduction in fat mass compared to the no EMS group ($p < 0.001$), and a significant increase in muscle mass compared to the no EMS group ($p = 0.002$).

## Biochemical characteristics of subjects before, at 4 weeks, and after 8 weeks of EMS intervention

Table 3 presents the biochemical characteristics of the subjects before the EMS intervention, at 4 weeks, and after the 8-week intervention period. Blood samples were collected every 4 weeks to analyze liver function, kidney function, and other indicators. It was observed that regardless of the EMS intervention, the blood biochemical indicators related to liver and kidney function did not show any abnormalities.

**Table 2 Effect of 8 weeks EMS combined with RT on body composition.**

| Body composition | No EMS | Daily EMS |
|---|---|---|
| Pre | | |
| Body weight (kg) | 64.2 ± 14.4 | 68.5 ± 15.5 |
| BMI (kg/m$^2$) | 22.3 ± 2.8 | 24 ± 3.2 |
| Muscle mass (kg) | 28.8 ± 7.2 | 29.5 ± 7.3 |
| Fat mass (%) | 20.3 ± 5.3 | 22.1 ± 6.4 |
| Post | | |
| Body weight (kg) | 66.5 ± 14.2[*] | 66.9 ± 15.2[*] |
| BMI (kg/m$^2$) | 23.1 ± 2.6[*] | 23.5 ± 3.1[*] |
| Muscle mass (kg) | 28.6 ± 7.1 | 30.3 ± 7.5[*] |
| Fat mass (%) | 22.7 ± 4.3[*] | 21.2 ± 7.1[*] |
| Delta post-pre | | |
| Muscle mass (kg) | −0.2 ± 0.9[a] | 0.8 ± 0.4[b] |
| Fat mass (%) | 2.4 ± 1.9[b] | −1.0 ± 1.3[a] |

Notes:
Data are presented as mean ± SD. Different superscript letters (a, b) indicate significant difference between groups at $p < 0.05$.
[*] The intervention efficacy between pre- and post-intervention within the same group was statistically significant at a significance level of $p < 0.05$. BMI, body mass index.

**Table 3 Biochemical characteristics of subjects before, 4-week and after 8-week EMS intervention.**

| Parameters | No EMS | Daily EMS |
|---|---|---|
| AST (U/L) | | |
| Pre | 20 ± 3 | 21 ± 3 |
| Mid | 19 ± 3 | 19 ± 2 |
| Post | 20 ± 3 | 20 ± 2 |
| ALT (U/L) | | |
| Pre | 20 ± 3 | 20 ± 3 |
| Mid | 21 ± 3 | 20 ± 3 |
| Post | 21 ± 2 | 20 ± 3 |
| BUN (mg/dL) | | |
| Pre | 15.7 ± 2.4 | 15.9 ± 2.7 |
| Mid | 15.7 ± 2.5 | 15.9 ± 2.6 |
| Post | 16.0 ± 2.5 | 16.1 ± 2.2 |
| CREA (mg/dL) | | |
| Pre | 1.09 ± 0.10 | 1.10 ± 0.07 |
| Mid | 1.13 ± 0.06 | 1.10 ± 0.06 |
| Post | 1.11 ± 0.08 | 1.10 ± 0.07 |
| UA (mg/dL) | | |
| Pre | 4.9 ± 0.9 | 5.0 ± 1.2 |
| Mid | 4.8 ± 0.9 | 5.0 ± 1.1 |
| Post | 4.8 ± 1.0 | 5.0 ± 1.1 |
| TP (mg/dL) | | |

(Continued)

| Table 3 (continued) | | |
|---|---|---|
| Parameters | No EMS | Daily EMS |
| Pre | 7.3 ± 0.3 | 7.3 ± 0.5 |
| Mid | 7.4 ± 0.3 | 7.3 ± 0.2 |
| Post | 7.3 ± 0.2 | 7.3 ± 0.4 |
| FFA (mmol/L) | | |
| Pre | 0.45 ± 0.17 | 0.46 ± 0.19 |
| Mid | 0.41 ± 0.10 | 0.41 ± 0.14 |
| Post | 0.41 ± 0.10 | 0.41 ± 0.14 |

Note:
Data are presented as mean ± SD. AST, aspartate aminotransferase; ALT, alanine transaminase; BUN, blood urea nitrogen; CREA, creatinine; UA, uric acid; FFA, free fatty acid.

Table 4 Effect of 8 weeks EMS combined with RT on physiological adaptation and biochemical indices.

| Fatigue index | No EMS | Daily EMS |
|---|---|---|
| Lactate (mmol/L) | | |
| E0′ | 1.46 ± 0.34 | 1.44 ± 0.28 |
| E30′ | 3.40 ± 0.55 | 3.21 ± 0.66 |
| R60′ | 1.48 ± 0.3 | 1.49 ± 0.23 |
| $NH_3$ (μmol/L) | | |
| E0′ | 64.2 ± 8.5 | 65.7 ± 10.8 |
| E30′ | 92.1 ± 5.9 | 92.3 ± 5.8 |
| R60′ | 66.3 ± 8.7 | 67.7 ± 10.9 |
| Glucose (mg/dL) | | |
| E0′ | 86 ± 7 | 88 ± |
| E30′ | 87 ± 7 | 91 ± 6 |
| R60 | 88 ± 8 | 87 ± 7 |

Note:
Data are presented as mean ± SD.

## Effect of 8 weeks EMS combined with RT on physiological adaptation and biochemical indices

As shown in Table 4, serum lactate and $NH_3$ levels increased with the prolongation of exercise time until reaching 30 min of exercise intervention, and gradually decreased to baseline levels during the recovery period. However, there were no significant differences between the no EMS group and the daily EMS group.

## Effect of 8 weeks EMS combined with RT on maximum handgrip strength

After 8 weeks of intervention, both the no-EMS group and the daily EMS group showed a significant increase in left-hand (no-EMS group, 1.03-fold ($p = 0.001$); daily EMS group 1.09-fold ($p < 0.001$)) and right-hand grip strength (no-EMS group, 1.04-fold ($p < 0.001$); daily EMS group 1.12-fold ($p < 0.001$)) compared to pre-intervention ($p < 0.05$) in within-group comparisons. However, there were no significant differences between the

**Table 5 Effect of 8 weeks EMS combined with RT on maximum handgrip strength.**

| Grip strength | No EMS | Daily EMS |
|---|---|---|
| Left (kg) | | |
| Pre | $34.4 \pm 10.4^a$ | $32.0 \pm 10.2^a$ |
| Post | $35.4 \pm 10.5^{a,*}$ | $34.7 \pm 9.8^{a,*}$ |
| Right (kg) | | |
| Pre | $36.2 \pm 11.5^a$ | $35.2 \pm 10.9^a$ |
| Post | $37.6 \pm 11.4^{a,*}$ | $39.5 \pm 11.0^{a,*}$ |
| Delta post-pre | | |
| Left (kg) | $1.0 \pm 0.9^a$ | $2.7 \pm 1.9^b$ |
| Right (kg) | $1.4 \pm 0.9^a$ | $4.4 \pm 2.8^b$ |

Notes:
Data are presented as mean ± SD. Different superscript letters (a, b) indicate significant difference between groups at $p < 0.05$.
*The intervention efficacy between pre- and post-intervention within the same group was statistically significant at a significance level of $p < 0.05$.

**Table 6 Effect of 8 weeks EMS combined with RT on muscle strength and jumping force.**

| CMJ | No EMS | Daily EMS |
|---|---|---|
| RFD (N/kg*sec) | | |
| Pre | $8.0 \pm 2.5$ | $8.0 \pm 1.0$ |
| Post | $9.0 \pm 2.3^*$ | $9.1 \pm 1.2^*$ |
| Peak force (N) | | |
| Pre | $974 \pm 325$ | $980 \pm 237$ |
| Post | $1,028 \pm 324^*$ | $1,048 \pm 244^*$ |
| Jump height (cm) | | |
| Pre | $27.3 \pm 4.6$ | $28.6 \pm 8.5$ |
| Post | $29.6 \pm 5.7^*$ | $31.0 \pm 8.4^*$ |
| Delta post-pre | | |
| RFD (N/kg*sec) | $0.9 \pm 0.9$ | $1.1 \pm 0.7$ |
| Peak force (N) | $54 \pm 37$ | $67 \pm 51$ |
| Jump height (cm) | $2.3 \pm 2.3$ | $2.4 \pm 2.1$ |

Notes:
Data are presented as mean ± SD.
*The intervention efficacy between pre- and post-intervention within the same group was statistically significant at a significance level of $p < 0.05$. RFD, rate of force development.

two groups. On exploring the incremental changes in left- or right-hand grip strength by calculating the pre- and post-intervention differences, we found that the daily EMS group had a significantly greater increase in left-hand ($p = 0.007$) and right-hand grip strength ($p = 0.002$) compared to the no EMS group (Table 5).

## Effect of 8 weeks EMS combined with RT on muscle strength and jumping force

The improvements in lower body strength and power were assessed using the CMJ test. As shown in Table 6, there were no significant differences between the no EMS and daily

EMS groups in terms of rate of force development (RFD), peak force, and jump height. After 8 weeks of intervention, although both groups showed significant improvements compared to before the intervention in within-group comparisons, there was no significant difference between the two groups in terms of the differences observed (*Lee et al., 2022*).

## DISCUSSION

In recent years, numerous studies have demonstrated the advantages of EMS as a passive exercise for muscles. However, the gains in muscle strength or mass may still be influenced by the specific protocols and frequency of use (*Mukherjee, Fok & van Mechelen, 2023*). In our current study, we observed that combining resistance training (RT) with EMS resulted in significant improvements in muscle mass and upper limb muscle strength, while there was only a non-significant trend for lower limb strength. Importantly, we found no evidence of abnormal liver and kidney function when EMS was used appropriately.

EMS is believed to have a significant impact on reducing body fat by increasing the resting metabolic rate (*Kemmler et al., 2016a*), leading to sustained reductions in body fat. This effect may be attributed to increased physical activity, which is associated with improved lipoprotein profiles and increased fat oxidation (*Kemmler et al., 2016b*, *2018a*). A systematic review of 23 articles demonstrated that the EMS group experienced significant improvements in muscle mass and function, as well as reductions in fat mass and lower back pain (*Kemmler et al., 2018b*). The simultaneous loss of fat and increase in muscle hypertrophy suggests that muscle hypertrophy may counteract the negative metabolic effects of intramuscular fat (*Gorgey & Shepherd, 2010*). However, there is still ongoing debate regarding the effects of EMS on fat loss, which may be due to variations in study populations and intervention protocols (*Kemmler et al., 2021*). Our study, similar to a previous study that combined 8 weeks of RT and EMS in untrained healthy adults and utilized the same BIA model for measuring body composition, observed a slight decrease in body fat mass in the RT combined with EMS group, while the RT alone group showed minimal change. However, these changes were not statistically significant in either within-group or between-group analyses (*Yoo et al., 2023*). In our study, involving healthy adults undergoing RT with EMS, we observed a significant 4.33% reduction in body fat percentage in the daily EMS group compared to baseline, and these reductions were also significant when compared to the no EMS group (Table 2).

While there are differing opinions on the effects of EMS on fat loss, its benefits for increasing muscle mass are widely recognized. The mechanisms underlying the muscle mass gain observed with EMS involve several factors. EMS has been shown to enhance the regenerative capacity of satellite cells fused with mature skeletal fibers (*Di Filippo et al., 2017*), promoting muscle regeneration through fibrosis-induced nuclear proliferation (*Cabric, Appell & Resic, 1987*). Additionally, EMS training increases cytoplasmic calcium concentrations, leading to alterations in muscle-specific transcriptional mechanisms and upregulation of myogenic transcription factors, such as D and G, in myogenic precursor cells, thereby promoting muscle mass improvements (*Teschler et al., 2021*). In the context of improving sarcopenia in the elderly, EMS has been found to promote the secretion of

myokines, particularly insulin-like growth factor-1 (IGF-1), and enhance downstream signaling pathways (*Leal, Lopes & Batista, 2018*; *Sajer, Guardiero & Scicchitano, 2018*). It also decreases the expression of ubiquitin ligase genes associated with muscular dystrophy, such as MuRF-1 and Atrogin-1 (*Kern et al., 2014*). EMS is also believed to enhance muscle mass and strength by reducing reactive oxygen species production, increasing overall skeletal muscle protein synthesis rates, and reducing catabolism (*Agergaard et al., 2017*). Therefore, EMS exhibits similar benefits to traditional strength training in terms of improving muscle mass. However, the physiological effects and adaptations to muscle activation patterns in the neuromuscular system may differ between EMS and voluntary contractions (*Yoo et al., 2023*). During voluntary contractions, motor unit recruitment follows the size principle, with small to large motor units being activated in a physiological manner. In contrast, EMS applies external current to stimulate axonal fibers, preferentially activating large motor units with low impedance (*Henneman, Somjen & Carpenter, 1965*). EMS is more likely to directly activate muscles beneath the stimulating electrodes compared to conventional exercise. Consequently, EMS combined with RT can enhance motor innervation, particularly the coordination between agonist and antagonist muscles, resulting in improved skeletal muscle physiological responses and neuromuscular adaptations through increased motor unit recruitment (*Paillard, 2008*; *Ogasawara et al., 2013*). Previous studies have demonstrated that EMS training 1.5 times per week can improve muscle mass, while training 1.5–3 times per week can significantly increase lean body mass (*Kemmler et al., 2016a, 2021*). In our study, after 8 weeks of continuous EMS combined with RT intervention, significant improvements in muscle mass were observed within and between the groups (Table 2). However, further research is still needed to elucidate the specific mechanisms involved.

Muscle mass and strength have been found to be closely correlated (*Chen et al., 2013*). Plyometric training has been shown to enhance the expression of fast-twitch myosin heavy chain isoforms, which can contribute to increased muscle force generation. The proportion of muscle fiber types also influences the muscle's ability to generate force, and targeting a higher percentage of type II fibers compared to type I fibers can lead to improvements in overall muscle strength and performance (*Rahmati, Gondin & Malakoutinia, 2021*; *Hasan et al., 2022*). EMS has been reported to enhance the force-generating capacity of muscles by preferentially stimulating adaptations in type II fibers, promoting actin-myosin bridging with calcium (*Kaplan et al., 2002*). In a previous study, we demonstrated that the addition of EMS to plyometric training resulted in significant improvements in strength and physical performance immediately and after 8 weeks of intervention (*Bouguetoch, Martin & Grosprêtre, 2021*). Grip strength is considered a reliable indicator of overall muscle strength (*Chan et al., 2022*). In our current study, we found that EMS combined with resistance exercise training not only improved muscle mass but also significantly increased maximum grip strength in both hands (Table 5). However, we did not observe significant changes in lower extremity strength (Table 6), suggesting the need for further exploration to determine whether different muscle types require specific EMS protocols to elicit significant improvements.

Excessive accumulation of fatigue by-products, such as lactic acid and $NH_3$, is believed to contribute to a decline in exercise performance as exercise duration or intensity increases (*Halson et al., 2002*). Lactic acid is produced during intense exercise as a result of carbohydrate breakdown *via* glycolysis under anaerobic conditions. The release of hydrogen ions during this process inhibits glycolysis and lowers the pH of blood and muscle tissue, which can interfere with normal cellular functions (*Proia et al., 2016*). Ammonia, on the other hand, is produced during amino acid metabolism for ATP energy production and can accumulate in skeletal muscle, affecting central fatigue (*Chen et al., 2020*). Previous studies have reported that incorporating EMS during exercise can recruit high-threshold motor units and muscle fibers, potentially enhancing performance (*Sawada et al., 2022*). In our study, we performed a fixed-intensity fatigue test after 8 weeks of RT combined with EMS to evaluate whether daily EMS training had benefits in improving fatigue threshold, reducing fatigue production, and accelerating recovery. However, based on the results presented in Table 4, no significant improvement in fatigue was observed (Table 4). Nevertheless, it is worth noting that the current study confirmed that 8 weeks of EMS use by healthy adults did not cause liver or kidney damage. While EMS is believed to enhance lipid metabolism and increase the utilization of energy substrates compared to voluntary muscle contraction, leading to increased concentrations of free fatty acids (*Hioki et al., 2023*), we did not observe any changes in free fatty acid concentrations in this study (Table 6).

Although this study demonstrated that 8 weeks of resistance exercise training combined with EMS improves muscle mass and strength in young, healthy adults, there are several limitations that should be acknowledged. Firstly, the study only included a specific population of young, healthy subjects, limiting the generalizability of the findings to other populations. Additionally, the study focused on the short-term effects of EMS during the 8-week intervention period and did not assess the long-term effects of EMS on muscle and physical performance when used independently. Secondly, the sample size of the study was relatively small, which, although consistent with the calculated effective sample size, may have had a slight impact on the results. Therefore, future studies with larger sample sizes should be conducted to confirm these findings. Thirdly, although the study aimed to explore differences in body composition and muscle strength changes between different EMS, the inclusion of both male and female subjects in the same group for comparison may have been a limitation, as strength and physiological differences between genders can significantly influence the results. Fourthly, while bioelectrical impedance analysis (BIA) is a practical method for assessing body composition, it provides a global assessment and may benefit from the addition of local body composition analysis using dual-energy X-ray absorptiometry. Moreover, the study did not implement standardized interventions targeting nutrient intake. Although participants were instructed to maintain their usual eating habits during the trial, there was limited monitoring of dietary adherence. These limitations highlight areas for improvement and suggest the need for future research with more diverse populations, longer follow-up periods, larger sample sizes, and the inclusion of standardized interventions and more precise body composition assessment methods. Furthermore, in-depth mechanism research is warranted to explore the factors

contributing to the effects of different EMS training programs on muscle mass and strength performance.

## CONCLUSIONS

Our study demonstrates that 8 weeks of continuous resistance exercise training combined with daily EMS training targeting the upper body, lower body, and abdominal muscles can significantly improve muscle mass and upper body muscle strength, as well as significantly reduce body fat percentage. However, it did not show a significant effect on lower body explosive force. These findings suggest that the combination of resistance exercise training and EMS can be an effective strategy for enhancing muscle mass and upper body strength while reducing body fat percentage. In this study, we used a multi-frequency cyclic transformation method to perform EMS, giving different levels of muscle stimulation under 30 min of training. Our design philosophy is that this may have a larger effect than a fixed frequency. However, we need to further compare the advantages of different EMS solutions. In addition, further research is needed to explore the potential mechanisms underlying these effects and to optimize the protocols for EMS training in order to maximize its benefits on muscle performance.

### Funding
This study was funded by the University–Industry Cooperation Fund, National Taiwan Sport University, Taoyuan, Taiwan (NTSU No. 1101062). The funders had no role in study design, data collection and analysis, decision to publish, or preparation of the manuscript.

### Grant Disclosures
The following grant information was disclosed by the authors:
University–Industry Cooperation Fund, National Taiwan Sport University, Taoyuan, Taiwan: 1101062.

### Competing Interests
The authors declare that they have no competing interests.

### Author Contributions
- Mon-Chien Lee conceived and designed the experiments, performed the experiments, analyzed the data, prepared figures and/or tables, authored or reviewed drafts of the article, and approved the final draft.
- Chin-Shan Ho performed the experiments, analyzed the data, prepared figures and/or tables, and approved the final draft.
- Yi-Ju Hsu performed the experiments, analyzed the data, prepared figures and/or tables, and approved the final draft.
- Ming-Fang Wu conceived and designed the experiments, authored or reviewed drafts of the article, materials, and approved the final draft.

● Chi-Chang Huang conceived and designed the experiments, prepared figures and/or tables, authored or reviewed drafts of the article, and approved the final draft.

## Human Ethics

The following information was supplied relating to ethical approvals (*i.e.*, approving body and any reference numbers):

Institutional Review Board of Landseed International Hospital (Taoyuan, Taiwan; LSHIRB No. 21-034-A2).

## Data Availability

All the data are available in the Supplemental File.

## Supplemental Information

Supplemental information for this article can be found online at http://dx.doi.org/10.7717/peerj.16303#supplemental-information.

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
