# Peer review of "Effect of 8-week frequency-specific electrical muscle stimulation combined with resistance exercise training on muscle mass, strength, and body composition in men and women: a feasibility and safety study"

_PeerJ, doi:10.7717/peerj.16303_

## Round 0.1 · original submission · Major Revisions

Dear Authors,
The reviewers and I have completed our evaluation of your manuscript and recommend a major revision before re-submission.
Please review the comments and resubmit your revised manuscript.

Reviewer 1 ·

Basic reporting

Abstract

Line 24-26: Describe the sample (age, weight, BMI, etc).
Line 31-33: Describe the exact p-values.
Detail in the abstract what type of population the study corresponds to (healthy subjects, physically inactive, etc).

Introduction:

Line 42-47: update the bibliographic citations.
The introduction is very long and makes it difficult to read, I suggest the authors reduce the size of the paragraphs.

Experimental design

Materials and methods:

Line 121: Was the clinical trial registered? Does it have a registration code?
Line 137: Is the device valid and reliable? What is the sampling frequency of the device? What other technical characteristics does it have?
Line 152: What was the criterion for progressing intensity in the maximal strength protocol.
Line 199: The authors mention that the CMJ is a reliable test, which bibliographic reference supports this statement?

Validity of the findings

Results

Please place the exact p-values and to 3 decimal places.

Table 1
What was the progress criterion in the EMS training protocol?

·

Basic reporting

The publication is written in good English. The structure is clear and background information with reference to EMS and strength training is given. The aim, methods, and the study design are precisely described.

Experimental design

The experimental design is relevant and meaningful. The methods and tests are well described and details are attached. It is interesting that the EMS group trains daily. In many publcations the number of training units per week ist clearly restricted to fewer input (1-2 training units/week). Can you explain why daily EMS training was performed?

Validity of the findings

A lot of tests were perfomed and all data for both investigated groups were offered and statistically sound interpreted. In the discussion, the results were discussed with reference to many other publications in the field of EMS and strength training.
The conclusion could be more precise. It is not clear how following studies shall be designed in order to optimize the outcome. Training frequency, loading, combination of EMS and strength methods etc. are missing.

Additional comments

The publication is clearly structured and the results and conclusions are understandable. Please explain the design of daily training of the EMS group and be more precise on the conclusions when offering study designs of new studies.

---

## Round 0.2 · accepted · Accept

Your manuscript has been accepted for publication. Congratulations!

·

Basic reporting

2nd revision: The authors followed the advice of the comments. So from my point of view the paper is ready for publication.

Experimental design

2nd revision: The authors followed the advice of the comments. So from my point of view the paper is ready for publication.

Validity of the findings

2nd revision: The authors followed the advice of the comments. So from my point of view the paper is ready for publication.

Additional comments

2nd revision: The authors followed the advice of the comments. So from my point of view the paper is ready for publication.